# Efficacy of Aceclofenac and Ilaprazole Combination Therapy versus Celecoxib Monotherapy for Treating NSAID-Induced Dyspepsia in Lumbar Spinal Stenosis Patients

**DOI:** 10.3390/medicina59071307

**Published:** 2023-07-14

**Authors:** Sanghoon Lee, Jung Guel Kim, Ho-Joong Kim

**Affiliations:** Department of Orthopedic Surgery, Seoul National University Bundang Hospital, Seoul National University College of Medicine, Seongnam-si 13620, Republic of Korea; sanghoon92@gmail.com (S.L.); franklin14172@gmail.com (J.G.K.)

**Keywords:** lumbar spinal stenosis 1, dyspepsia 2, proton pump inhibitor 3, NSAID 4, llaprazole 5

## Abstract

*Background and Objectives*: Dyspepsia is a common adverse event associated with the use of nonsteroidal anti-inflammatory drugs (NSAIDs) in patients with lumbar spinal stenosis. Although proton pump and cyclooxygenase-2 inhibitors are potential treatment options, the optimal strategy remains unclear. This study aimed to compare the efficacy and safety of combination therapy with aceclofenac and ilaprazole versus celecoxib monotherapy for the treatment of dyspepsia caused by NSAID use in patients with lumbar spinal stenosis. *Materials and Methods*: This prospective, double-blind, randomized, actively controlled study was conducted at Seoul National University Bundang Hospital in South Korea from July 2020 to September 2021. The participants were randomized into one of two treatment groups: celecoxib monotherapy (control group) and combination therapy with aceclofenac and ilaprazole (test group). The primary efficacy endpoint was the mean change in the Short-Form Leeds Dyspepsia Questionnaire (SF-LDQ) scores from baseline to treatment week 8. The secondary efficacy endpoint was the mean change in Short-Form-12 (SF-12) scores from baseline (week 0) to treatment week 8. *Results*: The study enrolled 140 patients who were randomly assigned to receive combination therapy with aceclofenac and, ilaprazole or celecoxib. In the per protocol set, the mean change in SF-LDQ scores from week 0 to week 8 was −0.51 ± 4.78 and 1.85 ± 6.70 in the combination therapy and celecoxib group, respectively (*p* = 0.054). SF-12 scores did not differ significantly between the two groups. Adverse events were reported in both groups, but there was no significant difference in incidence. *Conclusions*: Combination therapy with aceclofenac and ilaprazole can be a treatment option for NSAID-induced dyspepsia in some situations.

## 1. Introduction

Nonsteroidal anti-inflammatory drugs (NSAIDs) are widely used for pain relief and inflammation management in patients with lumbar spinal stenosis [1,2]. The main mechanism of action is the inhibition of the enzyme cyclooxygenase (COX). Cyclooxygenase is required to covert arachidonic acid into thromboxanes, prostaglandins, and prostacyclins. The therapeutic effects of NSAIDs are attributed to the lack of these eicosanoids. However, their use can carry adverse gastrointestinal events, including dyspepsia, peptic ulcers, and bleeding [3]. While dyspepsia can occur for various reasons, the use of NSAIDs is one of its most common causes. Several treatment options are available to manage NSAID-induced dyspepsia, including proton pump inhibitors (PPIs), histamine-2 receptor antagonists (H2RAs), and other gastroprotective agents [4,5]. However, the optimal treatment strategy for NSAID-induced dyspepsia remains unclear.

One potential treatment strategy for this condition is to switch from traditional NSAIDs to celecoxib, a cyclooxygenase-2 selective inhibitor that effectively manages pain and inflammation and is associated with a lower risk of gastrointestinal adverse events [6,7].

Alternatively, combining NSAIDs with PPIs could be an option. Aceclofenac is another nonsteroidal anti-inflammatory drug that is widely used to manage pain and inflammation, particularly in conditions such as osteoarthritis, rheumatoid arthritis, and lumbar spinal stenosis [8]. It is a phenylacetic acid derivative that is structurally related to diclofenac, which showed a higher therapeutic index than other NSAIDs. In addition, aceclofenac showed reduced gastrointestinal AEs compared to other traditional NSAIDs. Ilaprazole, on the other hand, is a new PPI that belongs to the category of substituted benzimidazole molecules; it is chemically related to omeprazole. The mechanism of action is similar between ilaprazole and omeprazole, in which the benzimidazoles suppress gastric acid secretion by inhibiting the H+/K+-ATPase on the secretory surfaces of gastric parietal cells [9]. Through this mechanism, they effectively manage dyspepsia and other acid-related disorders [10]. Therefore, the combination of aceclofenac and ilaprazole is a potential treatment option for patients with NSAID-induced dyspepsia. However, studies on which treatment option is better for this condition are lacking.

Therefore, this study aimed to evaluate the efficacy and safety of aceclofenac and ilaprazole combination therapy versus celecoxib monotherapy for the treatment of NSAID-induced dyspepsia.

## 2. Materials and Methods

### 2.1. Study Design

This prospective, double-blind, randomized, actively controlled study was conducted at Seoul National University Bundang Hospital in South Korea from July 2020 to September 2021. This study was approved by the Institutional Review Board of Seoul National University Bundang Hospital. Informed consent was obtained from all patients. All participants and clinicians were blinded to the settings, which consisted of week 0 (visit one, screening visit), week 4 (visit two, by telephone), and week 8 (visit three, end of study) visits. Participants who met the inclusion criteria were randomized into one of two treatment groups at a 1:1 ratio: celecoxib monotherapy (control group, 200 mg twice daily) or combination therapy with aceclofenac and ilaprazole (test group, aceclofenac 100 mg twice daily, ilaprazole 10 mg once daily). Web-based randomization was employed to achieve concealed allocation. All participants were evaluated for drug efficacy and safety over the 8-week study period.

### 2.2. Particpants

The inclusion criteria were as follows: patients aged ≤80 years who voluntarily agreed to participate and who were taking nonselective NSAIDs continuously due to lumbar spinal stenosis. The exclusion criteria were as follows: (1) patients with a previous diagnosis of esophageal stenosis, ulcer stenosis, esophageal gastric varices, Barrett esophagus, active peptic ulcer, gastrointestinal bleeding, or malignant tumor; (2) patients with hypersensitivity to the test drug or any component of the test drug or any drug of the same class; (3) patients who used H2 receptor antagonists, prostaglandin drugs, mucosal protection agents, or prokinetics within the previous 4 weeks; (4) patients who used of any type of NSAID, including aspirin, steroids, or anticoagulants; (5) patients with congestive heart failure (New York Heart Association class II–IV) or established ischemic heart disease, peripheral arterial disease, or cerebrovascular disease; (6) patients with hyperkalemia; (7) patients with a presence of a bleeding or coagulation disorder; (8) patients with chronic liver disease or renal impairment; (9) patients with a history of gastric acid secretion suppression surgery or gastric/duodenal surgery; and (10) patients who were deemed unfit for participation in this clinical trial by the researcher.

### 2.3. Efficacy and Safety Evaluation

The primary efficacy endpoint was the mean change in the Short-Form Leeds Dyspepsia Questionnaire (SF-LDQ) scores from baseline to treatment week 8. The SF-LDQ is a shortened version of the Leeds dyspepsia questionnaire, a tool used to assess the severity and impact of dyspepsia on an individual’s daily life [11]. The LDQ consists of eight questions that pertain to dyspeptic symptoms, along with an additional question regarding the patient’s most troublesome symptom. contains eight questions relating to dyspeptic symptoms, and one question about the most troublesome symptom for the patient. The SF-LDQ, on the other hand, comprises the four questions from the LDQ that have the greatest validity with respect to dyspepsia. Every question consists of two parts that specifically address the frequency and severity of each symptom experienced by the patient over the past two months. Additionally, the SF-LDQ includes an additional question that inquires about the patient’s most troublesome symptom. comprises two stems focused on the frequency and severity of each symptom during the last 2 months. The SF-LDQ also contains a single question concerning the most troublesome symptom experienced by the patient. A high score indicates more severe dyspepsia symptoms. The questionnaire used in this study originated from Fraser’s study and has been validated for use in Korean [11]. We translated the SF-LDQ questionnaire into Korean using an authorized translation method.

The secondary efficacy endpoint was the mean change in Short-Form-12 (SF-12) scores from baseline to treatment week 8. The SF-12 is a shortened version of the 36-item Short-Form Health Survey, which is a widely used tool for assessing health-related quality of life [12]. The SF-12, a self-report questionnaire that assesses an individual’s physical and mental health status, generates two summary scores: the physical component summary (PCS) and the mental component summary (MCS) scores. It uses the same eight domains as the SF-36, as follows: (1) limitations in physical activities because of health problems; (2) limitations in social activities because of physical or emotional problems; (3) limitations in usual role activities because of physical health problems; (4) bodily pain; (5) general mental health (psychological distress and well-being); (6) limitations in usual role activities because of emotional problems; (7) vitality (energy and fatigue); and (8) general health perceptions. A higher score indicates a better quality of life. In addition, for safety analysis, adverse events, clinical laboratory tests, vital signs, and physical examination results were assessed. Serious adverse events refer to the following situations that occur while taking clinical drugs: (1) death or a life-threatening condition; (2) hospitalization or the need for an extended hospitalization period; and (3) continuous or significant disability or deterioration.

### 2.4. Sample Size Calculation

The sample size required to evaluate the superiority of combination therapy over celecoxib monotherapy was calculated via the equation below, using the following conditions: Z_α/2_ = 1.96, Z_β_ = 0.84, *d* = 0.43–0.1 = 0.33, σ = 0.66 [13]:$$N=\frac{\left(Z_{\alpha /2}+Z_{\beta }\right)\times 2}{d/\sigma }$$

A sample size of 63 patients per group was determined using the above formula. Subsequently, considering a dropout rate of 10%, a final sample size of 70 patients per group was obtained.

### 2.5. Statistical Analysis

Data from this study were analyzed in safety, full analysis (FA), and per protocol (PP) sets. The safety set included all patients who received at least one drug during the study. The FA set included all patients for whom primary efficacy endpoints were obtained at least once before the end of the study. The PP set comprised patients who completed the clinical trial without violating the study protocol. Efficacy evaluation data were obtained from the FA and PP sets. Safety evaluation data were obtained from the safety set.

All statistical analyses were performed using SPSS (version 27.0; IBM Corp., Armonk, NY, USA). Continuous variables were expressed as mean and standard deviation, while categorical variables were expressed as number and percentage. A two-sided test with a significance level of 5% was used for all analyses. A repeated-measures analysis of variance (ANOVA) was conducted to analyze the primary efficacy endpoint. The within-subject factor was time, with three levels. The independent sample *t*-test was used to compare continuous variables between the two groups, and the chi-squared test was used to categorize the variables.

## 3. Results

### 3.1. Demographic Characteristics

A total of 140 patients was enrolled in this study (Figure 1). One patient did not meet the inclusion/exclusion criteria and 11 patients withdrew consent. The remaining 128 patients were randomly assigned to receive combination therapy of aceclofenac and ilaprazole (64 patients) or celecoxib therapy (64 patients). Of the 128 patients, 103 completed the clinical trial according to the protocol (combination therapy group, 51; celecoxib group, 52). The FA set consisted of all 125 patients for whom the data on primary efficacy variable were obtained. The PP set (n = 103) excluded 22 patients with visit-schedule violations (n = 6 in the combination therapy group and n = 4 in the celecoxib group) or adverse events (n = 6 in the combination therapy group and n = 6 in the celecoxib group).

The patients’ demographic characteristics are shown in Table 1. There were 80 (62.5%) women and 48 men (37.5%) aged 45–80 (mean age, 67.6) and 40–86 (mean age, 67.2) years in the combination and celecoxib groups, respectively. There were no statistically significant intergroup differences in terms of sex, age, height, weight, or body mass index. The patients’ baseline dyspepsia scores are shown in Table 1. The mean baseline SF-LDQ score was 3.03 ± 4.57 in the combination therapy group and 2.91 ± 4.19 in the celecoxib group. The mean baseline PCS and MCS, components of the SF-12, were 39.01 ± 7.34 and 46.89 ± 9.81 in the combination therapy group and 38.54 ± 8.83 and 48.36 ± 10.70 in the celecoxib group, respectively. The differences between the two groups were not statistically significant with respect to the baseline characteristics of dyspepsia.

### 3.2. Primary Efficacy Results

The primary efficacy results are presented in Table 2 and Figure 2. In the PP set, the SF-LDQ score of the combination therapy group was 2.86 at week 0, 2.76 at week 4, and 1.94 at week 8. The mean change in SF-LDQ scores from week 0 to week 8 was −0.51 ± 4.78. In the celecoxib group, the mean SF-LDQ score was 2.42 at week 0, 3.15 at week 4, and 4.27 at week 8. The mean change in the SF-LDQ scores was 1.85 ± 6.70. The repeated-measures ANOVA showed a significant interaction between time and group (*p* = 0.047), indicating that the effect of treatment on the SF-LDQ scores changed over time. SF-LDQ decreased only in the aceclofenac and ilaprazole groups after 8 weeks. However, the intergroup difference in SF-LDQ scores from baseline to week 8 was not statistically significant (*p* = 0.054). In addition, the results of the FA set were similar to those of the PP set. In the FA set, the SF-LDQ score of the combination therapy group was 2.76 at week 0, 2.84 at week 4, and 2.88 at week 8. The mean change in SF-LDQ scores from week 0 to week 8 was 0.13 ± 5.30. In the celecoxib group, the mean SF-LDQ score was 2.89 at week 0, 4.61 at week 4, and 4.27 at week 8. The mean change in the SF-LDQ scores was 1.85 ± 6.70. The intergroup difference in SF-LDQ scores from baseline to week 8 was not significant in the PP set (*p* = 0.140).

### 3.3. Secondary Efficacy Results

The secondary efficacy results are shown in Table 3. In the PP set, the PCS/MCS scores of the combination therapy group were 38.99/47.48 at week 0, 41.36/48.35 at week 4, and 42.09/48.98 at week 8. The mean change in PCS/MCS scores from week 0 to 8 was 3.10/1.49. In the celecoxib group, the PCS/MCS scores were 39.03/49.83 at week 0, 41.01/50.38 at week 4, and 40.19/50.20 at week 8. The mean change in the PCS/MCS scores was 1.16/0.37. There were no significant differences in the mean change in the PCS/MCS scores from week 0 to week 8 (*p* = 0.310 and 0.597, respectively). The results for the FA set were similar to those for the PP set. In the FA set, the PCS/MCS scores of the combination therapy group were 39.01/47.04 at week 0, 41.05/47.47 at week 4, and 41.13/48.81 at week 8. The mean change in PCS/MCS scores from week 0 to 8 was 2.54/1.51. In the celecoxib group, the PCS/MCS scores were 38.71/48.34 at week 0, 41.13/48.57 at week 4, and 40.19/50.20 at week 8. The mean change in the PCS/MCS scores was 1.16/0.37. There were no significant differences in the mean change in the PCS/MCS scores from week 0 to week 8 in the PP set (*p* = 0.457 and 0.589, respectively).

### 3.4. Safety Results

Adverse events (AEs) were reported in seven (10.9%) and six (9.4%) patients in the combination therapy and celecoxib groups, respectively, with no significant difference in their incidence (*p* = 0.770) (Table 4). No serious AEs resulting in life-threatening conditions or severe sequelae were observed.

## 4. Discussion

This randomized double-blind clinical trial was the first to compare the efficacy of a combination therapy of aceclofenac plus ilaprazole with celecoxib monotherapy for NSAID-induced dyspepsia. This study showed no significant intergroup difference in treatment efficacy or safety. Although there were no statistically significant differences in the outcome values, meaningful results were found for the clinical application of ilaprazole.

Analysis of the PP set revealed that, while celecoxib monotherapy resulted in deterioration, combination therapy exhibited improved SF-LDQ scores at week 8 (*p* = 0.054), which was borderline significant. Based on these findings, combination therapy with aceclofenac and ilaprazole may be effective in the treatment of NSAID-induced dyspepsia. In particular, the difference in the SF-LDQ scores increased by week 8, suggesting that the efficacy of ilaprazole in NSAID-induced dyspepsia may be even greater when administered for a longer duration. Therefore, patients who experience dyspepsia and require long-term NSAID treatment may benefit from combination treatment with aceclofenac and ilaprazole.

As mentioned previously, celecoxib may be a good option for patients with NSAID-induced dyspepsia [6,7]. However, in recent studies, cyclooxygenase-2 selective inhibitors, including celecoxib, have been associated with increased cardiovascular events [14,15,16,17,18]. In addition, it was demonstrated that celecoxib use was associated with a dose-related increase in the composite and point of death from cardiovascular causes, myocardial infarction, stroke, or heart failure. Evidence that selective inhibition of COX-2 can prevent the generation of prostacyclin without impacting the synthesis of thromboxane A2, perhaps resulting in a prothrombotic state, suggests one possible mechanism for this effect [19]. Considering this disadvantage, in patients with a previous history of cardiovascular disease, celecoxib is typically not used. In such cases, combination therapy with aceclofenac, a nonselective NSAID, and ilaprazole could be a safe and viable alternative, since no significant differences in complication rates were noted between the two treatment options.

The efficacy of ilaprazole for treating duodenal ulcers was demonstrated in several studies [20,21,22]. Wang et al. demonstrated that the ulcer healing rates were 93.0% in the ilaprazole treatment group, and the majority of patients became a symptomatic after 4 weeks of treatment [9]. Ilaprazole is highly effective and safe compared with other PPIs in the treatment of duodenal ulcer. However, no research has been conducted on its ability to prevent NSAID-induced gastrointestinal problems. This study showed that ilaprazole could effectively prevent gastrointestinal problems in patients using NSAIDs, but further studies are required to examine its efficacy at preventing NSAID-induced ulcers.

The safety of ilaprazole has been demonstrated in several studies. Wang et al. conducted a study comparing ilaprazole doses of 5, 10, or 20 mg/day with a 20 mg/day dose of omeprazole, and they showed that ilaprazole was comparable to omeprazole and there were no clinically relevant changes in the hematology and biochemistry test results [9]. Another clinical trial showed that oral doses of ilaprazole—10, 20 or 40 mg once daily—were well-tolerated and considered safe in healthy subjects [23]. Similarly, this clinical trial demonstrated that the combination of ilaprazole with NSAIDs was a safe option. However, further studies are needed to determine whether ilaprazole, when administered in combination with NSAIDs, is safer compared to existing PPIs, such as omeprazole.

This study had several limitations. First, despite planning a superiority test and setting a specific number of participants, no significant intergroup differences were found. A non-inferior clinical trial may be necessary. Second, the final number of study participants was small because of the high rate of scheduled-visit violations and loss of follow-up visits. However, as originally planned, the FA set included a sufficient number of patients. Third, most patients with symptomatic osteoarthritis typically use NSAIDs for a prolonged period. However, the follow-up duration in this study was only 8 weeks. Patients with osteoarthritis take NSAIDs for an extended period, necessitating a longer study period.

## 5. Conclusions

Our findings suggest that, in some cases, a combination of aceclofenac and ilaprazole can be an alternative to celecoxib for treating NSAID-induced dyspepsia. However, further research, such as non-inferior clinical trials, is required to determine its efficacy.

## Figures and Tables

**Figure 1 medicina-59-01307-f001:**
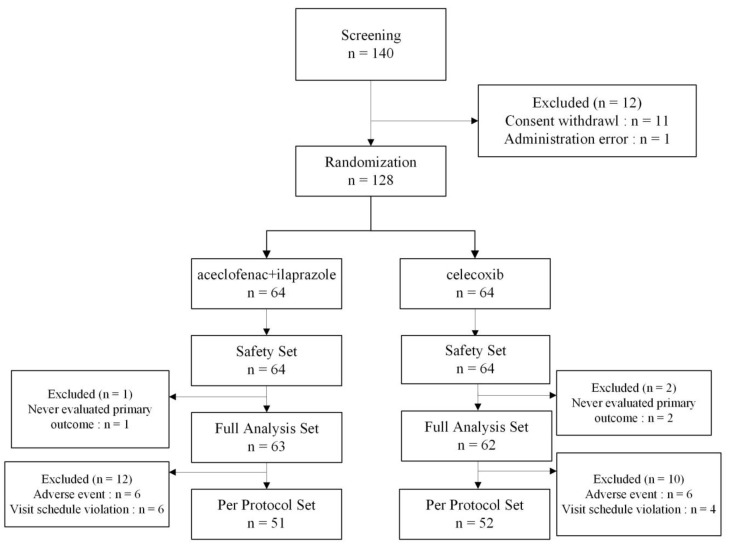
Flowchart of this randomized clinical trial.

**Figure 2 medicina-59-01307-f002:**
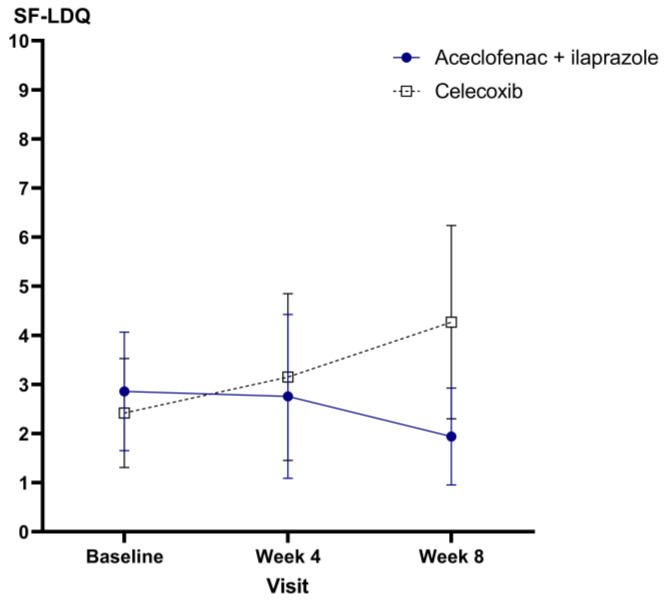
Mean changes with 95% confidence intervals in Short-Form Leeds Dyspepsia Questionnaire scores throughout the study period.

**Table 1 medicina-59-01307-t001:** Demographic and baseline characteristics.

	Aceclofenac + Ilaprazole n = 64	Celecoxib n = 64	*p*-Value
Age (years)	67.6 ± 8.4	67.2 ± 8.3	0.792
Sex (No.)			0.465
Female	42 (65.6%)	38 (59.4%)	
Male	22 (34.4%)	26 (40.6%)	
Height, cm	156.35 ± 8.35	158.01 ± 8.54	0.237
Weight, kg	61.97 ± 9.53	63.52 ± 9.75	0.410
SF-LDQ ^a^ (initial)	3.03 ± 4.57	2.91 ± 4.19	0.872
SF-12 ^b^ (initial)			
PCS ^c^	39.01 ± 7.34	38.54 ± 8.83	0.694
MCS ^d^	46.89 ± 9.81	48.36 ± 10.70	0.421

^a^ Short-Form Leeds Dyspepsia Questionnaire; ^b^ Korea short-form health survey; ^c^ physical component summary; ^d^ mental component summary.

**Table 2 medicina-59-01307-t002:** Changes in SF-LDQ ^a^ from baseline to Week 8.

	Aceclofenac + Ilaprazole	Celecoxib	*p*-Value
Full analysis set			
Baseline	2.76 ± 4.06	2.89 ± 4.21	0.866
Week 4	2.84 ± 5.71	4.61 ± 8.06	0.162
Week 8	2.88 ± 4.93	4.27 ± 7.07	0.234
Difference (Week 8—Baseline)	0.13 ± 5.30	1.85 ± 6.70	0.140
Per protocol set			
Baseline	2.86 ± 4.29	2.42 ± 3.99	0.580
Week 4	2.76 ± 5.93	3.15 ± 6.10	0.493
Week 8	1.94 ± 3.51	4.27 ± 7.07	0.286
Difference (Week 8—Baseline)	−0.51 ± 4.78	1.85 ± 6.70	0.054

^a^ Short form-Leeds dyspepsia questionnaire.

**Table 3 medicina-59-01307-t003:** Changes in secondary outcome values from baseline to Week 8.

	Aceclofenac + Ilaprazole	Celecoxib	*p*-Value
Full analysis set			
PCS ^a^			
Baseline	39.01 ± 7.36	38.71 ± 8.91	
Week 4	41.05 ± 7.78	41.13 ± 9.22	
Week 8	41.13 ± 9.34	40.19 ± 9.39	
Difference (Week 8—Baseline)	2.54 ± 8.34	1.16 ± 10.78	0.457
MCS ^b^			
Baseline	47.04 ± 9.82	48.34 ± 10.84	
Week 4	47.47 ± 10.08	48.57 ± 10.42	
Week 8	48.81 ± 11.06	50.20 ± 10.32	
Difference (Week 8—Baseline)	1.51 ± 12.10	0.37 ± 9.55	0.589
Per protocol set			
PCS ^a^			
Baseline	38.99 ± 7.08	39.03 ± 9.43	
Week 4	41.36 ± 7.62	41.01 ± 9.65	
Week 8	42.09 ± 9.21	40.19 ± 9.39	
Difference (Week 8—Baseline)	3.10 ± 8.38	1.16 ± 10.78	0.310
MCS ^b^			
Baseline	47.48 ± 9.73	49.83 ± 10.43	
Week 4	48.35 ± 9.45	50.38 ± 9.50	
Week 8	48.98 ± 11.08	50.20 ± 10.32	
Difference (Week 8—Baseline)	1.49 ± 11.92	0.37 ± 9.55	0.597

^a^ physical component summary; ^b^ mental component summary.

**Table 4 medicina-59-01307-t004:** Adverse events based on the safety analysis set.

	Aceclofenac + Ilaprazole n = 64	Celecoxib n = 64	*p*-Value
Incidence of adverse events	7 (10.9%)	6 (9.4%)	0.770
GI trouble	3	4	
Swelling	3	0	
Redness	0	1	
Dehydration	1	1	

## Data Availability

Data can be obtained from the corresponding author upon reasonable request.

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
