# Peer review of "Efficacy of Aceclofenac and Ilaprazole Combination Therapy versus Celecoxib Monotherapy for Treating NSAID-Induced Dyspepsia in Lumbar Spinal Stenosis Patients"

_medicina, 2023, doi:10.3390/medicina59071307_

Round 1
Reviewer 1 Report
In the present study, the authors aimed to describe the efficacy of two pharmacological therapies for NSAID-induced dyspepsia, when administered for lumbar spinal stenosis. The efficacy of the two treatments was determined by using two questionnaires to assess the presence and severity of dyspepsia ("Short-Form Leeds Dyspepsia Questionnaire") and the patients` quality of life „Short-Form-12”.
The results of the present study are interesting and may serve to improve the management of NSAID-induced dyspepsia. Because of the small number of patients, the results were not necessarily significant, but as the authors noted, perhaps in future studies, on a larger number of patients, the results will be more conclusive.
Authors should mention in their study whether the questionnaires were validated in their native language and describe the validation process.
Author Response
Authors should mention in their study whether the questionnaires were validated in their native language and describe the validation process.:
Thank you for you thoughtful comment. The questionnaire used in this study originated from Fraser’s study and has been validated for use in Korean. We have added this in the "2.3 Efficacy and safety evaluation".
Reviewer 2 Report
This study showed that ilaprazole could effectively prevent gastrointestinal problems in patients using NSAIDs, but further studies are required to examine its efficacy at preventing NSAID-induced ulcers.
This study had several limitations. First, despite planning a superiority test and setting a specific number of participants, no significant intergroup differences were found. A non-inferior clinical trial may be necessary. Second, the final number of study participants was small because of the high rate of scheduled-visit violations and loss to followup. However, as originally planned, the FA set included a sufficient number of patients. Third, most patients with symptomatic osteoarthritis typically use NSAIDs for a prolonged period. However, the follow-up duration in this study was only 8 weeks. Patients with osteoarthritis take NSAIDs for an extended period, necessitating a longer study period.

Author Response
Further studies are required to examine its efficacy at preventing NSAID-induced ulcers.
-> Thank you for your thoughtful comment. We have added your comment in the discussion section. Thank you.
This study had several limitations.
-> Thank you for your thoughtful comment. We have mentioned about your comment in the discussion section. Thank you.
Round 2
Reviewer 1 Report
I thank the authors for their answers to my comments. The statement “The questionnaire used in this study originated from Fraser’s study and has been validated for use in Korean” was added. I recommend the authors to add the reference for the questionnaire validation in Korean if they have it, if not, at least to resume the translation and validation process. Please consider the same recommendations for the second questionnaire (Short-Form-12).
Also, it is important to paraphrase “The LDQ contained eight questions relating to dyspeptic symptoms, and one question about the most troublesome symptom for the patient. The SF-LDQ contained the four questions from the LDQ which had the greatest validity compared with dyspepsia. Each question comprised two stems about the frequency and severity of each symptom during the last 2 months. The SF-LDQ also contained a single question concerning the most troublesome symptom experienced by the patient.”, as it was identically taken from the original article, reference 9.